# Acetylation Enhances the Anticancer Activity and Oral Bioavailability of 5-Demethyltangeretin

**DOI:** 10.3390/ijms232113284

**Published:** 2022-10-31

**Authors:** Hui-Yun Tsai, Jia-Fang Yang, Yin-Bo Chen, Jia-Lin Guo, Shiming Li, Guor-Jien Wei, Chi-Tang Ho, Jue-Liang Hsu, Chi-I Chang, Yu-Shen Liang, Hsu-Sheng Yu, Yu-Kuo Chen

**Affiliations:** 1Department of Nutrition and Health Science, Fooyin University, Kaohsiung 83102, Taiwan; 2Aging and Disease Prevention Research Center, Fooyin University, Kaohsiung 83102, Taiwan; 3Department of Food Science, National Pingtung University of Science and Technology, Pingtung 91201, Taiwan; 4Department of Food Science, Rutgers University, New Brunswick, NJ 08901, USA; 5Institute of Food Safety and Health Risk Assessment, National Yang Ming Chiao Tung University, Taipei 112304, Taiwan; 6Department of Biological Science and Technology, National Pingtung University of Science and Technology, Pingtung 91201, Taiwan; 7Department of Plant Industry, National Pingtung University of Science and Technology, Pingtung 91201, Taiwan

**Keywords:** anticancer, citrus, polymethoxyflavones, acetylation, bioavailability

## Abstract

A kind of hydroxylated polymethoxyflavone (PMFs) existing in the citrus genus, 5-Demethyltangeretin (5-DTAN), has been reported to possess several bioactivities in vitro and in vivo. The aim of this study was to investigate whether acetylation could enhance the anticancer activity and oral bioavailability of 5-DTAN. PC-3 human prostate cancer cells were treated with tangeretin (TAN), 5-DTAN, and 5-acetylated TAN (5-ATAN), and the results showed that the cytotoxic effect 5-ATAN (IC_50_ value of 5.1 µM) on the cell viability of PC-3 cells was stronger than that of TAN (IC_50_ value of 17.2 µM) and 5-DTAN (IC_50_ value of 11.8 µM). Compared to 5-DTAN, 5-ATAN treatment caused a more pronounced DNA ladder, increased the sub-G1 phase population, and induced G2/M phase arrest in the cell cycle of PC-3 cells. We also found that 5-ATAN triggered the activation of caspase-3 and the progression of the intrinsic mitochondrial pathway in PC-3 cells, suggesting the induction of apoptosis. In a cell wound healing test, 5-ATAN dose-dependently reduced the cell migration, and the expression of matrix metalloproteinase-2 (MMP-2) and matrix metalloproteinase-9 (MMP-9) was decreased after 48 h of 5-ATAN treatment. Moreover, oral administration of 5-ATAN showed a significantly stronger inhibitory effect on tumor size and tumor weight in tumor-bearing nude mice than those of vehicle or the 5-DTAN group (*p* < 0.05). Furthermore, pharmacokinetic results showed that single-dose oral administration of 5-ATAN exhibited a higher maximum concentration (Cmax) and area under the curve (AUC) of 5-DTAN in plasma than that of 5-DTAN. More extensive distribution of 5-DTAN to most tissues of mice was also observed in mice treated with 5-ATAN for 7 days. In conclusion, acetylation strongly enhances the anticancer activity and oral bioavailability of 5-DTAN and could be a promising strategy to promote the potential bioactivities of natural products.

## 1. Introduction

Polymethoxyflavonoids (PMFs), flavonoids containing multiple methoxy groups in the structure, are a class of compounds widely present in citrus plants of the Rutaceae family, especially in the peel of sweet oranges (*Citrus sinensis*) and mandarin oranges (*C. reticulate*). To date, dozens of PMFs have been isolated and identified from citrus plants, and their types and contents are different with the type of citrus species [1]. Tangeretin (5,6,7,8,4′-pentamethoxyflavone, TAN), one of the most abundant PMFs in citrus peel, has been reported to possess several biological activities, including anti-oxidative, anti-inflammatory, hepatoprotective, neuroprotective, anti-obesity, and anticancer effects [2,3,4,5,6,7]. Certain studies in previous years have shown that 5-hydroxylated PMFs formed during storage by the autohydrolysis of PMFs in citrus fruits exhibited more potent bioactivities than their PMF counterparts [8,9,10]. Among them, 5-demethyltangeretin (5-hydroxy-6,7,8,4′-tetramethoxyflavone, 5-DTAN), a structural analog of TAN, exerted higher activities than its counterpart TAN in inhibiting the growth and angiogenesis of cancer cells [11,12]. Moreover, 5-DTAN effectively suppressed the growth of human non-small cell lung cancer (NSCLC) cells by arresting the cell cycle in the G2/M phase, and its effect is markedly stronger than TAN [13]. A study reported by Ma et al. [14] also demonstrated that 5-DTAN was more potent than TAN in regulating phosphoinositide 3-kinases (PI3K), cyclooxygenase-2 (COX-2), and AKT expression in skin cancer cells, thereby reducing skin cancer cell growth and tumorigenesis. However, the extremely poor solubility of 5-DTAN in biological systems may limit its bioavailability and bioavailability [15]. 

Prodrugs are deliberately designed as derivatives of drug molecules that can be converted to active parent drugs by biocatalytic or chemical transformation in biological systems [16]. Chemical modification, one of the most common methods to produce prodrugs, is performed by adding another moiety, most commonly an ester, to introduce lipophilicity and bury the hydrogen-bonding groups of the parent drugs [17]. Adding esters to the drugs commonly can enhance the oral bioavailability and minimize the toxicity of the drugs [18]. For instance, aspirin is an acetic acid ester derivative of salicylic acid by acetylation [19]; morphine is acetylated with acetic anhydride to produce heroin [20]. In addition, Lambert et al. [21] pointed out that the absorption and anticancer activity of EGCG were significantly increased after the treatment of human esophageal cell line KYSE150 and colorectal cancer cell line HCT116 with acetylated EGCG. A study conducted on the pharmacokinetic tests of resveratrol and its acetylated derivative, 3,5,4’-tri-O-acetylresveratrol (TARES), showed that the area under the curve (AUC) value of TARES is higher than that of resveratrol, indicating that acetylated resveratrol has a higher absorption in rats, which can improve the therapeutic effects of resveratrol on cardiovascular and respiratory diseases [22].

Wang et al. [23] obtained the acetyl derivatives of TAN (5-acetyloxy-6,7,8,4′-tetramethoxyflavone, 5-ATAN) by demethylation and the acetylation modification of TAN and demonstrated that 5-ATAN effectively caused cytotoxicity to MCF-7 human breast cancer cells through the induction of apoptosis. Briefly, 5-ATAN also has been found to induce the mitochondria-associated apoptotic pathway and exert a pro-apoptotic effect on human multiple myeloma U226 cells [24]. Furthermore, our previous study showed that 5-ATAN induced cell cycle G2/M arrest, apoptosis, and autophagy in CL1-5 human non-small cell lung cancer cells in vitro and in vivo [25]. It has also been reported that 5-ATAN treatment markedly suppressed the cell growth and colony formation capacity of human glioblastoma multiforme cells via the blockade of the signal transducer and activator of the transcription 3 (STAT3) signaling pathway [26]. However, to date, no study has proven the anticancer effects of 5-ATAN against human prostate cancer cells, and the underlying mechanism, as well as bioavailability, needs to be elucidated.

In the present study, we investigated the inhibitory effects of 5-ATAN on the cell growth of the PC-3 human prostate cancer cell line in vitro and in vivo. Moreover, the cellular uptake of de-esterified 5-ATAN in vitro and its oral bioavailability in mice, as well as tissue distribution, were also examined.

## 2. Results and Discussion

### 2.1. Effects of TAN, 5-DTAN, and 5-ATAN on Cell Proliferation of PC-3 Human Prostate Cancer Cells 

Figure 1A shows the chemical structures of TAN, 5-DTAN, and 5-ATAN. The MTT assay was performed to compare the effects of these compounds on the cell proliferation of PC-3 cells. The cell viability was determined after cells were exposed to various concentrations of TAN, 5-DTAN, and 5-ATAN for 48 h. From Figure 1B, it can be seen that all three PMFs used in this experiment significantly inhibited the cell viability of PC-3 cells in a dose–response manner, and the IC_50_ values of TAN, 5-DTAN, and 5-ATAN were 17.2 µM, 11.8 µM, and 5.1 µM, respectively. Briefly, 5-ATAN showed a higher inhibitory effect, and that its IC_50_ value was 3.4-fold and 2.3-fold lower than that of TAN and 5-DTAN, respectively. Edible agricultural products have been reported to contain a variety of phytochemicals with biological activities. TAN is one of the most studied and abundant compounds among the PMFs present in the peels of citrus. In the past decade, the efforts of research have been focused on its 5-hydroxylated derivative, 5-DTAN, because 5-DTAN showed stronger activity against carcinogenesis than its PMF counterpart, TAN [13,14], which was also observed in the present study. Despite its higher anticancer activity, the application of 5-DTAN is limited due to its poor solubility in biological systems [15]. We obtained 5-ATAN, which is the product of acetylation at the C5 position of 5-DTAN, and investigated whether 5-ATAN impacts the cell growth of PC-3 human prostate cancer cells. From the above results, it is convincing that 5-ATAN possessed much stronger inhibitory effects on PC-3 cells than TAN and 5-DTAN.

### 2.2. The 5-ATAN-Induced Cell Cycle Arrest and Apoptosis in PC-3 Human Prostate Cancer Cells

We further examined the mechanism of 5-ATAN against PC-3 cells using DNA fragmentation assay and cell cycle analysis. The results showed that significant DAN ladders were observed in the treatment of 5-ATAN at concentrations higher than 5 μM, indicating the occurrence of apoptosis (Figure 2A). However, 5-DTAN caused obvious DNA laddering only at the highest treatment concentration (20 μM). Moreover, the treatment of 5-ATAN resulted in a higher apoptotic ratio in the cell cycle of PC-3 cells compared with that of 5-DTAN (Figure 2B), and treatment with 10 μM of 5-ATAN caused 22.2% apoptotic cells in PC-3 cells. Wang et al. [23] measured the DNA damage induced by 5-ATAN or TAN by single cell gel electrophoresis (SCGE) and found that 5-ATAN treatment induced a higher increase in DNA breakage in MCF-7 human breast cancer cells compared with TAN. After the PC-3 cells were treated with 5-ATAN, DNA fragmentation occurred in the nucleus, which represented the occurrence of apoptosis. In general, when cells undergo apoptosis, in addition to cell shrinkage, chromatin condensation, and the production of apoptotic bodies, the activation of caspase-3 also prompts caspase-activated DNase (CAD) to degrade DNA. The DNA was cleaved into fragments with multiples of 180–200 bp, resulting in DNA fragmentation [27,28]. The fragmentation of nuclear DNA also can be detected by the discrete sub-G1 phase on DNA content histograms analyzed by flow cytometry [29,30]. Briefly, 5-ATAN treatment significantly increased the ratio of the sub-G1 phase of the cell cycle in a dose-dependent manner, suggesting that 5-ATAN promoted PC-3 cells toward apoptosis. Furthermore, 5-ATAN treatment also affected the progression of the cell cycle in PC-3 cells. As is shown in Table 1, cells treated with 5-ATAN resulted in arrest in the G2/M phase, accompanied by a reduced distribution in the G0/G1 phase, whereas cells treated with 5-DTAN were less pronounced. Similarly, Xiao et al. [31] reported that 5-demethylated PMFs inhibited the growth of H1299 human lung cancer cells through apoptosis but did not affect the cell cycle at the same treatment concentration. Furthermore, our previous study showed that G2/M cell cycle arrest and apoptosis also participated in the reaction of CL1-5 NSCLC cells to 5-ATAN [25].

### 2.3. Effects of 5-ATAN on Intrinsic Mitochondrial Pathway and Cell Cycle Checkpoint Proteins in PC-3 Human Prostate Cancer Cells

Apoptosis can be divided into three major pathways, but no matter which pathway it goes through, it will eventually achieve the purpose of cell disintegration through the activation of caspase-3 [32]. In the present study, PC-3 cells were treated with different concentrations of 5-ATAN (2.5, 5, and 10 µM) for 48 h to investigate the effect of 5-ATAN on caspase-3 activity. The results showed that 5-ATAN dose-dependently increased caspase-3 activity in PC-3 cells, and the percentages of the caspase-3 activity of PC-3 cells treated with 5-ATAN were 7.2 ± 0.6%, 9.1 ± 1.7%, and 18.8 ± 02.4%, respectively (Figure 3A,C). To elucidate whether the intrinsic mitochondrial pathway is involved in 5-ATAN-induced apoptosis in PC-3 cells, we determined the mitochondrial membrane potential and the protein expression of Bcl-xL and Bax. From Figure 3B,D, it can be seen that 5-ATAN caused a dose-dependent decline of mitochondrial membrane potential in PC-3 cells. Moreover, the results of Western blot showed that 10 µM of 5-ATAN treatment resulted in a marked decrease in the expression of Bcl-xL, while a slight increase in Bax (Figure 3E) indicated that mitochondrial pathway-triggered apoptosis plays an important role in the inhibition of PC-3 cell growth by 5-ATAN. The above results are similar to those found by Zhi et al. [24], that 5-ATAN induced apoptosis in multiple myeloma U266 cells through the intrinsic mitochondrial pathway. We further analyzed certain checkpoint proteins associated with the regulation of the G2/M phase to clarify the mechanism of cell cycle arrest caused by 5-ATAN. The results showed that 5-ATAN dramatically down-regulated the expression of both Cdc25c and Cyclin B1 in a time-dependent manner, whereas Cdc-2 was only reduced after being treated with 5-ATAN for 48 h. Cdc25c plays a vital role in the activation of the Cdc2/Cyclin B1 complex, which is required for promoting cell cycle progression from the G2 to the M phase [33,34]. One of the previous studies reported by Li et al. [25] mentioned that 5-ATAN arrested the cell cycle in the G2/M phase in CL1-5 cells by the down-regulation of Cdc25c protein expression. Consequently, our results revealed that the suppression of the Cdc25C/Cdc2/Cyclin B pathway, at least in part, is involved in the 5-ATAN-induced growth inhibition of PC-3 cells via G2/M cell cycle arrest.

### 2.4. The 5-ATAN Inhibited Colony Formation and Cell migration in PC-3 Human Prostate Cancer Cells

A colony formation assay was performed to evaluate the effects of 5-ATAN on the growth ability of PC-3 cells. PC-3 cells were incubated with 1.25, 2.5, and 5 µM of 5-ATAN for 14 days. As shown in Figure 4A, the number of colonies of PC-3 cells dose-dependently decreased after treatment with 5-ATAN. Colony formation or clonogenic assay is an in vitro tumorigenic assay based on the capacity of a single cell to form a large colony of 50 or more cells [35,36]. Our findings from the colony formation assay indicated that 5-ATAN may possess potential antitumor effects in in vitro studies and even in vivo studies. We further examined the effects of 5-ATAN on the ability of cell migration in PC-3 cells via wound healing assay. PC-3 cells were subjected to the treatment with various concentrations of 5-ATAN for 24 h. The 5-ATAN exhibited significant and dose-dependent suppression on the cell migration (*p* < 0.05), and 5-ATAN at 1.25, 2.5, and 5 µM suppressed the migration of PC-3 cells by 35.5%, 48.5%, and 73.6% compared to the control, respectively (Figure 4B,C). Cell migration is a key procedure participating in morphogenesis, inflammation, and metastasis in cancer cells [37]. The spread of tumor cells is invariably the main cause of morbidity and mortality in cancer patients; therefore, most studies have focused on tumor cell migration as the underlying cause of cancer metastasis [38]. Wound healing assay, a common, low-cost, and extremely reproducible method, can be applied to measure cell motility in cancer cells, which has been identified as required for tumor cell metastasis. It was found that 5-ATAN significantly inhibited the cell motility of PC-3 cells, thereby reducing the cell migration ratio, indicating that 5-ATAN may have the ability of the inhibition of cancer metastasis. Figure 4D illustrates the effect of 5-ATAN on the expression of matrix metalloproteinase-2 (MMP-2) and matrix metalloproteinase-9 (MMP-9) proteins in PC-3 cells. Briefly, 5-ATAN significantly inhibited the expression of MMP-9 under the treatment concentration of 5 µM, and the inhibitory effect enhanced with the increase in treatment time. However, the effect of 5-ATAN on MMP-2 was less effective, and the expression of MMP-2 was obviously inhibited only at the 48 h mark of the treatment. In the process of tumor metastasis, wound healing, and angiogenesis, the MMP family of enzymes is responsible for the degradation of the extracellular matrix (ECM) and the regulation of cell invasion and migration [39,40]. Among the MMPs, MMP-2 and MMP-9 are typically involved in cancer cell invasion, bone tissue degradation, and angiogenesis due to their contribution to the degradation of the main component of the basement membrane [41]. Although MMP-2 and MMP-9 possess similar structures and catalytic activities, only MMP-9 can be activated by cytokines and growth factors in certain animal and cell models. Our results reveal that 5-ATAN time-dependently inhibited the expression of MMP-9, suggesting that 5-ATAN has the potential to reduce cancer cell migration, invasion, and metastasis.

### 2.5. Effects of 5-DTAN and 5-ATAN on Tumor Size and Weight of PC-3 Cell Xenografts in Nude Mice

The mouse xenograft model was conducted to evaluate the anti-tumor effects of 5-DTAN and 5-ATAN on the growth of PC-3 cells in vivo. We found that daily treatment of 10 mg/kg of 5-DTAN and 5-ATAN significantly suppressed the tumor volume from PC-3 cell xenografts in nude mice, and 5-ATAN showed a stronger inhibitory ability to tumor growth than that of 5-DTAN (*p* < 0.05, Figure 5A–C). At the end of the experiment, the tumor weight of the 5-ATAN treatment group was significantly lower than that of the 5-DTAN treatment and control groups (*p* < 0.05, Figure 5D), and their tumor weights are 0.94 ± 0.18 g, 1.39 ± 0.36 g, and 2.47 ± 0.43 g, respectively. The tumor growth inhibitory rates of the 5-DTAN and 5-ATAN groups, estimated by the means of tumor weight, were 43.7% and 61.9%, respectively. Wang et al. [42] extracted and purified nobiletin (NOB), TAN, 5-demethylnobiletin (5-DNOB), and other PMFs from Ougan (*Citrus reticulata* cv. *Suavissima*), an important citrus cultivar in China, and found that TAN showed an in vivo tumor-inhibitory effect on SGC-7901 human gastric cancer cells via the induction of apoptosis and the up-regulation of the RARβ protein in a nude mice tumor xenograft model. Our previous studies also indicated that 20 mg/kg of 5-ATAN administration (i.p.) significantly suppressed tumor growth with a 43% inhibitory rate in CL1-5 cell xenograft mice [25]. In the present study, we demonstrate that acetylation enhances the antitumor activity of 5-ATAN in PC-3 cell xenograft mice compared with its parent compound 5-DTAN. To our knowledge, this study is the first to compare the anticancer effects of 5-DTAN and 5-ATAN on PC-3 human prostate cancer cells and show that 5-ATAN reveals a stronger inhibitory ability than 5-DTAN.

### 2.6. Effects of Acetylation on the Cell Uptake of 5-DTAN in PC-3 Cells and Its Oral Bioavailability in C57BL/6J Mice

Previous reports have suggested that prodrugs derived by acetylation are rapidly converted to their parent drugs when transported in the blood or absorbed into cells [18,21]. For this reason, we further measured the cytosolic levels of 5-DTAN in 5-ATAN-treated PC-3 cells and the plasma contents of 5-DTAN in 5-ATAN-administered C57BL/6J mice to elucidate whether the enhanced bioactivities of 5-ATAN resulted from the increased cellular uptake ability and oral bioavailability of 5-DTAN. Figure 6A shows that the levels of 5-DTAN in the cytosol raised with the increased concentration of 5-ATAN treatment for 12 h in PC-3 cells, whereas those treated with 5-DTAN tended to remain after 10 µM treatment. At 15 µM treatment, the cytosolic concentration of 5-DTAN in 5-ATAN-treated cells was 5.8 ± 1.0 nmol/10^6^ cells and 2.3-fold higher than those treated with 5-DTAN, indicating that acetylation promoted the uptake capacity of 5-DTAN by PC-3 cells. Similarly, 5-ATAN increased intracellular 5-DTAN levels time-dependently and was about 4.9-fold more potent than 5-DTAN after 24 h treatment (Figure 6B). This may explain why the growth inhibitory effect of 5-ATAN was higher than that of 5-DTAN in PC-3 cells. A study investigated the effects of acetylated EGCG on KYSE150 human esophageal cells found that acetylation significantly increased the absorption and anticancer activity of EGCG [21], and this improvement may be achieved by the blockage of polar side-chains and the enhancement of hydrophobicity. Lam et al. [43] mentioned that acetylated EGCG, a prodrug of EGCG, showed enhanced inhibitory ability against the proteasome activity in intact cells rather than against pure enzymes, representing that deacetylation via cells is critical for the increase in the bioactivity of acetylated prodrugs. The pharmacokinetic assay was applied to investigate the oral bioavailability of 5-DTAN following a single i.g. 50 mg/kg of 5-DTAN or 5-ATAN administration since 5-ATAN was also rapidly converted back to 5-DTAN in the blood. Figure 6C illustrates representative chromatograms of standard stock, blank plasma, and plasma after the administration of 5-ATAN, and the 5-DTAN was eluted at about 12.8 min. We found that the plasma concentration of the 5-DTAN achieved a maximum (0.773 ± 0.196 μg/mL) about 60 min after administration in mice administered with 5-ATAN, whereas those treated with 5-DTAN had a maximal plasma concentration (0.030 ± 0.009 μg/mL) about 29 min after treatment (Figure 6D and Table 2). Although the t_1/2_ of the total 5-DTAN in the plasma of the 5-ATAN-treated mice was lower than that of the 5-DTAN-treated mice, the AUC of the total 5-DTAN enhanced by 11.0-fold in the plasma of mice administered with 5-ATAN compared with those administered with 5-DTAN. Our findings suggest that acetylation improves the oral bioavailability of 5-DTAN in mice. Liang et al. [22] investigated and compared the oral bioavailability of resveratrol and its acetylated derivative, 3,5,4’-tri-O-acetylresveratrol (TARES), via pharmacokinetic tests, and they found that the AUC value of TARES is greater than that of resveratrol, indicating that acetylated resveratrol has a higher absorption in rats. Moreover, acetylation enhanced the radical scavenging and anti-inflammatory activity of hyaluronic acid via its improved bioavailability in murine monocyte/macrophage cells [44].

### 2.7. Effects of Acetylation on Tissue Distribution of 5-DTAN in C57BL/6J Mice

Our results show that acetylation dramatically affects the plasma levels of 5-DTAN in mice treated with 5-ATAN. Therefore, we further examined and compared the tissue distribution of 5-DTAN following the administration of 50 mg/kg i.g. 5-DTAN or 5-ATAN to C57BL/6J mice for 7 consecutive days. The levels of 5-DTAN were measured in certain tissues including the liver, prostate, colon, spleen, kidney, lung, brain, and heart of mice, and the results are shown in Figure 7. At the end of the experiment, the levels of 5-DTAN in most tissues of mice treated with 5-ATAN were significantly higher than those of mice treated with 5-DTAN (*p* < 0.05), except for the kidney and lung. The above results indicate that acetylation promotes the oral bioavailability and tissue distribution of 5-DTAN in mice. Previous reports also found that acetylation enhanced the levels of EGCG and resveratrol accumulated in certain organs of mice or rats [21,22], but our study is the first to report the benefits of acetylation on the tissue distribution of 5-DTAN, or even of PMFs, in biological systems. Moreover, the concentrations of 5-DTAN in the prostate, pancreas, brain, and heart of 5-DTAN-treated mice are extremely low; however, comparable concentrations of 5-DTAN in those of 5-DTAN-treated mice can be detected. The considerable distribution in the prostate, pancreas, brain, and heart provided a promising therapeutic strategy using PMFs for certain diseases in these target organs, such as prostate cancer, diabetes, Alzheimer’s disease, and cardiovascular diseases, among others. In fact, our recent study indicated that acetylated NOB greatly improved cognitive impairment in mice with Alzheimer-like syndrome (not published yet).

## 3. Materials and Methods

### 3.1. Chemicals

Roswell Park Memorial Institute (RPMI) Medium 1640, fetal bovine serum (FBS), and penicillin-streptomycin (PS) were purchased from Gibco (Grand Island, NY, USA). Bovine serum albumin (BSA) and a Coomassie protein assay reagent kit were obtained from Thermo Fisher Scientific (Waltham, MA, USA). The Bax, Bcl-xL, Cyclin B1, and MMP-2 antibodies were purchased from Cell Signaling Technology (Danvers, MA, USA). The MMP-9 antibody was provided by Genetex (Irvine, CA, USA). The β-actin, Cdc2, and Cdc25c were purchased from Santa Cruz Biotechnology (Santa Cruz, CA, USA). The QIAamp DNA Mini Kit was obtained from Qiagen (Hilden, Germany). The 3-(4,5-dimethyl-2-thiazolyl)-2,5-diphenyl-2H-tetrazolium bromide (MTT), dimethyl sulfoxide (DMSO), Propidium iodode (PI), sodiumdodecyl sulfate (SDS), and all other chemicals were obtained from Sigma-Aldrich (St. Louis, MO, USA). TAN, 5-DTAN, and 5-ATAN were provided by Dr. Chi-Tang Ho (Department of Food Science, Rutgers University, New Brunswick, NJ, USA). 

### 3.2. Cell Culture and Treatments

PC-3 human prostate cancer cell line was obtained from the Bioresource Collection and Research Center (Hsinchu, Taiwan) and maintained in RPMI medium with 10% FBS and 1% penicillin-streptomycin (PS). The cells were incubated in a 5% CO_2_ humidified atmosphere at 37 °C. TAN, 5-DTAN, and 5-ATAN were dissolved in DMSO to prepare a 20 mM sample stock, and the final ratio of the DMSO in the treatment medium was 0.1%. 

### 3.3. Cell Viability, Cell Cycle, and DNA Fragmentation Analysis

PC-3 cells were seeded in 96-well plates at a density of 2.5 × 10^3^ cells/well and allowed to attach for 24 h. The medium was changed to serum-free RPMI supplemented with 0–20 μM TAN, 5-DTAN, or 5-ATAN, and then incubated for an additional 48 h. The cell viability was determined by the MTT assay and represented as a percentage of the growth of the DMSO-treated cells. FACScan flow cytometer (Becton Dickinson, San Jose, CA, USA) was used to analyze the cell cycle distribution as previously mentioned [45]. Briefly, PC-3 cells treated with various concentrations of 5-DTAN or 5-ATAN for 48 h were trypsinized and gently fixed in 70% ice-cold ethanol and stored at −20 °C for at least 24 h. Then, the cells were washed with cold phosphate-buffered saline (PBS), followed by incubation with 40 μg/mL PI, 0.1 mg/mL RNase A, and 0.1% TritonX-100 in the dark for 30 min at room temperature. Subsequently, the cell cycle distribution of the PI-stained cells was analyzed by a FACScan flow cytometer, and the results were analyzed and calculated using ModFit LT Ver. 3.0 software. The proportion of the sub-G1 phase was quantitated and represented as the apoptotic ratio. Furthermore, a QIAamp DNA mini kit was used to extract DNA from the PC-3 cells treated with 5-DTAN or 5-ATAN for 48 h. Then, the extracted DNA was loaded on a 2% agarose gel containing HealthView^TM^ Nucleic Acid Stain (Genomics, New Taipei, Taiwan) for electrophoresis at 50 V. The gel was visualized under UV light, and the DNA fragmentation pattern was photographed.

### 3.4. Caspase-3 Activity and Mitochondria Membrane Potential Analysis

The PC-3 cells were treated with the indicated concentration of 5-DTAN or 5-ATAN for 48 h, and then cells were collected and stained by a CaspGLOW^TM^ Fluorescein Active Caspase-3 Staining Kit (Thermo Fisher Scientific, MA, USA) or 3,3′-dihexyloxacarbocyanine iodide (DiOC6, Merck Millipore, Burlington, MA, USA) for caspase-3 activity and mitochondria membrane potential analysis, respectively. The percentages of caspase-3-activated or DiOC6-positive cells were measured using a flow cytometer and calculated by the CellQuest Pro software.

### 3.5. Colony Formation and Cell Migration Analysis

The colony formation assay was conducted using a procedure modified from the previous report [13]. PC-3 cells were seeded in 6-well plates at a density of 500 cells/well and allowed to attach for 24 h. Subsequently, the cells were treated with various concentrations of 5-ATAN. After 14 days of incubation, the colonies formed were observed by crystal violet staining and photographed. The wound healing assay was performed to assess the effect of 5-ATAN on the migration ability of PC-3 cells. Cells were seeded into a 6 cm dish at a cell density of 1 × 10^6^ cells/well. After being cultured at 37 °C for 24 h, a 200 µL yellow pipette tip was used to scrape a gap in the middle of the dish. After washing twice with 1X PBS solution, the medium was replaced with fresh, serum-free medium containing different concentrations of 5-ATAN and incubated for 24 h. Then, the images were taken with an inverted phase contrast microscope at a magnification of 100×, and the Image J software was used for the measurement of the wound width. The percentage of cell migration (% of control) was calculated by measuring the movement of the cells into the scratch area after 24 h of treatment with DMSO or 5-ATAN.

### 3.6. Protein Expression Analysis

Western blot was performed to analyze the protein expression in the PC-3 Cells. The cells were exposed to indicated concentrations of 5-ATAN for various time periods at 37 °C. Then, they were washed with cold PBS and harvested, and a Western blot was conducted as previously described [46,47].

### 3.7. Animals

Male nu/nu mice were obtained from BioLasco Taiwan (Taipei, Taiwan) and maintained on a standard laboratory chow (Purina rodent chow 5001, Purina, St. Louis, MO, USA). At 5 weeks of age, nude mice were used for the xenograft experiment. For the oral bioavailability study, 5-week-old male C54BL/6J mice were purchased from the National Laboratory Animal Center (Taipei, Taiwan), and the experiment was started after 1 week of acclimation. All mice were housed in an air-conditioned room (temperature: 22 ± 2 °C; humidity: 50–60%) under standard conditions (12 h light/12 h dark cycle). All animal studies were approved by the National Pingtung University of Science and Technology, Institutional Animal Care and Use Committee with protocol numbers NPUST-102-015 and NPUST-105-045.

### 3.8. Inhibition of Tumor Growth in the Xenograft Model

The 5-week-old nude mice were divided into three groups (*n* = 6) and administered with the vehicle, 5-DTAN (10 mg/kg), and 5-ATAN (10 mg/kg), respectively. Tumors were established by subcutaneous injections of PC-3 cells (1 × 10^6^ cells) into the right flank side of nude mice. The tumor size was measured twice a week with digital calipers, and the tumor volume (V) was calculated by the following formula: V = length × width^2^ × 0.5. On day 24, the mice were sacrificed by CO_2_, and the tumors were harvested and weighed.

### 3.9. Cellular Uptake, Oral Bioavailibiltiy, and Tissue Distribution Analysis

Basically, esterase in the cytosol and plasma rapidly converts 5-ATAN absorbed into cells or into the blood circulation to 5-DTAN by deacetylation. The 5-DTAN in 5-DTAN-treated and 5-ATAN-treated samples can be determined by the HPLC-electrochemical detection (ECD) method. PC-3 cells were seeded in 10 cm dishes (1 × 10^6^ cells) and allowed to attach for 24 h. Subsequently, cells were treated with various concentrations of 5-DTAN or 5-ATAN for 12 h, or with 15 µM of 5-DTAN or 5-ATAN for different time periods. Then, the cells were washed with cold PBS, collected, and centrifuged at 300× *g* for 5 min to remove the supernatant. A total of 150 μL ice PBS was added to suspend the cells and then it was sonicated. The supernatant was combined with an equal volume of methanol for extraction followed by centrifuging at 15,130× *g* for 15 min at 4 °C. The cytosolic levels of 5-DTAN in the resulting supernatant were analyzed by HPLC with an Ascentis RP-Amide column (L × I.D. 15 cm × 4.6 mm, 3 μm) (Merck KGaA, Darmstadt, Germany) as previously described [48]. Briefly, the mobile phase used in HPLC consisted of 40% acetonitrile, 10% tetrahydrofuran, 0.05% trifluoroacetic acid, and 50 mM ammonium acetate, and the flow rate was 1 mL/min. The analysis was conducted for 20 min, and the ECD detector voltage was set at 300 mV. For the oral bioavailability study, 6-week-old male C54BL/6J (*n* = 6) mice were orally administered a single dose of 5-DTAN or 5-ATAN (50 mg/kg) and collected plasma at 5, 10, 15, 30, 60, 90, 120, 150, 180, and 300 min. Briefly, 100 μL of plasma was extracted with a 5-fold volume of ethyl acetate twice, and the ethyl acetate fractions were pooled and dried under vacuum. The samples were reconstituted in methanol and the levels of 5-DTAN were analyzed by HPLC-ECD. Pharmacokinetic parameters, including maximum concentration (C_max_), maximum time (T_max_), half-life (T_1/2_), and area under the curve (AUC), were calculated and obtained by the Thermo kinetica 5.0 software (Thermo Fisher Scientific, Waltham, MA, USA). A test in which 5-DTAN or 5-ATAN (50 mg/kg) were given for 7 consecutive days was conducted to investigate the tissue distribution of 5-DTAN in C57BL/6J mice. At the end of the experiment, the tissues, including the liver, prostate, colon, spleen, kidney, lung, brain, and heart, were harvested and weighed. Subsequently, the tissue samples were homogenized in a 3-fold volume of cold saline using a Polytron disrupter, and the homogeneous solution was combined with 0.5 mL of ethyl acetate. The supernatant was obtained by centrifuging at 15,130× *g* for 15 min at 4 °C and then extracted with an equal volume of ethyl acetate and analyzed using the methods described above.

### 3.10. Statistical Analysis

Data are expressed as means ± standard deviation (SD). Statistical analysis of in vitro studies and comparisons of statistical significance between groups were determined by one-way analysis of variance (ANOVA) with Duncan’s test. Statistical analysis of in vivo studies and comparisons of statistically significant differences among groups were performed by a paired Student’s t-test. A *p* < 0.05 was considered statistically significant. 

## 4. Conclusions

To the best of our knowledge, this study is the first to systematically compare the anticancer activities of 5-DTAN and 5-ATAN and their oral bioavailability as well as tissue distribution in mice. We provide firm evidence that acetylation increased the cytotoxicity of 5-DTAN against PC-3 human prostate cancer cells through the induction of G2/M arrest, the production of mitochondria pathway-triggered apoptosis, and the enhancement of cellular uptake of 5-DTAN. Our results also indicate that the plasma levels and tissue distribution of 5-DTAN in C57BL/6J mice were significantly increased by acetylation, which can be a promising therapeutic strategy for treating the diseases of certain target organs with PMFs or their prodrugs. Overall, acetylation strongly enhances the anticancer activity and oral bioavailability of 5-DTAN and could be a potential tool to promote the bioactivities of natural components.

## Figures and Tables

**Figure 1 ijms-23-13284-f001:**
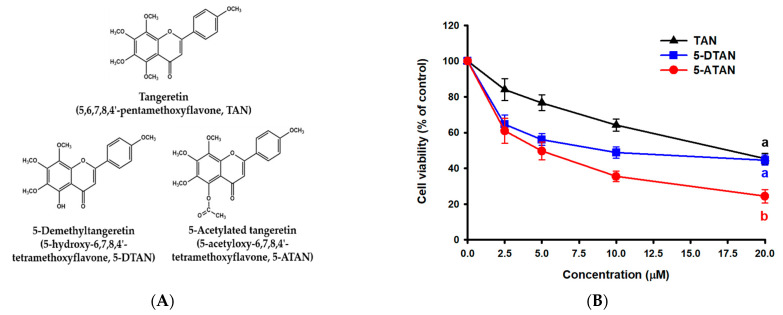
Effects of tangeretin (TAN), 5-demethyltangeretin (5-DTAN), and 5-acetylated tangeretin (5-ATAN) on cell viability in PC-3 cells. (**A**) Chemical structures of TAN, 5-DTAN, and 5-ATAN. (**B**) Cells were treated with various concentrations of TAN, 5-DTAN, or 5-ATAN for 48 h. Data are expressed as means ± SD (*n* = 3). The values with different letters are significantly different among treatments at the highest concentration (*p* < 0.05).

**Figure 2 ijms-23-13284-f002:**
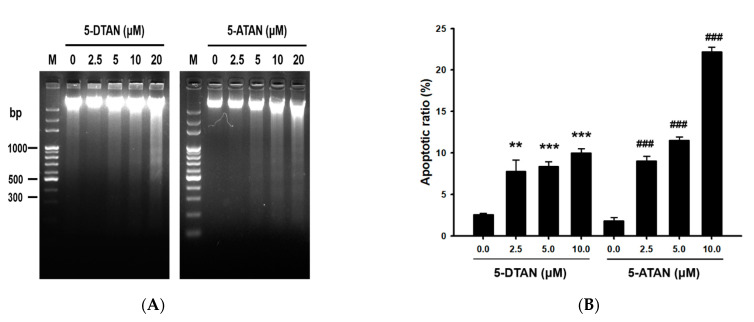
Effects of 5-DTAN and 5-ATAN on DNA fragmentation and apoptotic ratio in PC-3 cells. Cells were treated with various concentrations of 5-DTAN and 5-ATAN for 48 h. (**A**) DNA fragmentation was analyzed by electrophoresis in 2.0% agarose gel. (**B**) The apoptotic ratio of cells was calculated from the sub-G1 percentage of the cell cycle in PC-3 cells. Data are expressed as means ± SD (*n* = 3). ** *p*< 0.01, ***^,^ ### *p* < 0.001 vs. control group.

**Figure 3 ijms-23-13284-f003:**
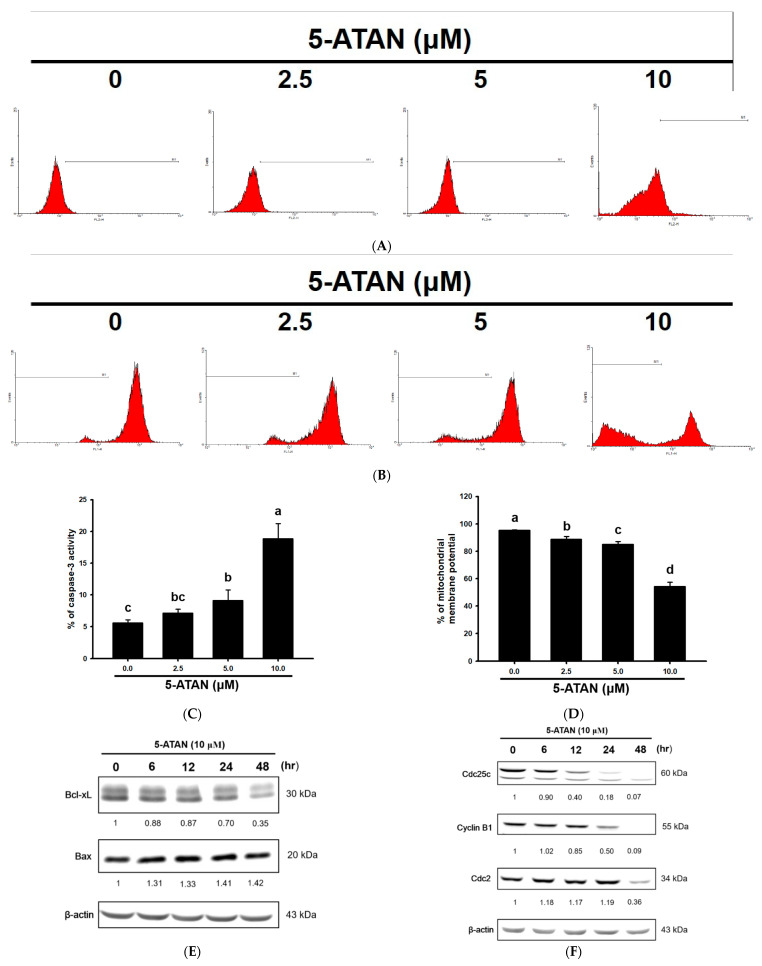
Effects of 5-ATAN on the mitochondrial pathway and cell cycle checkpoint protein expression in PC-3 cells. Cells were treated with various concentrations of 5-ATAN for 48 h or 10 µM of 5-ATAN for different time periods. (**A**) Caspase-3 activity. (**B**) Mitochondrial membrane potential was then analyzed by flow cytometry. (**C**,**D**) Data were expressed as mean ± SD of three determinations presented in the bar graph and analyzed statistically using ANOVA and Duncan’s test. Different letters represent statistical differences among treatments, *p* < 0.05. The protein expression of (**E**) Bcl-xL and Bax, (**F**) Cdc25c, Cyclin B1, and Cdc2 were analyzed by Western blot.

**Figure 4 ijms-23-13284-f004:**
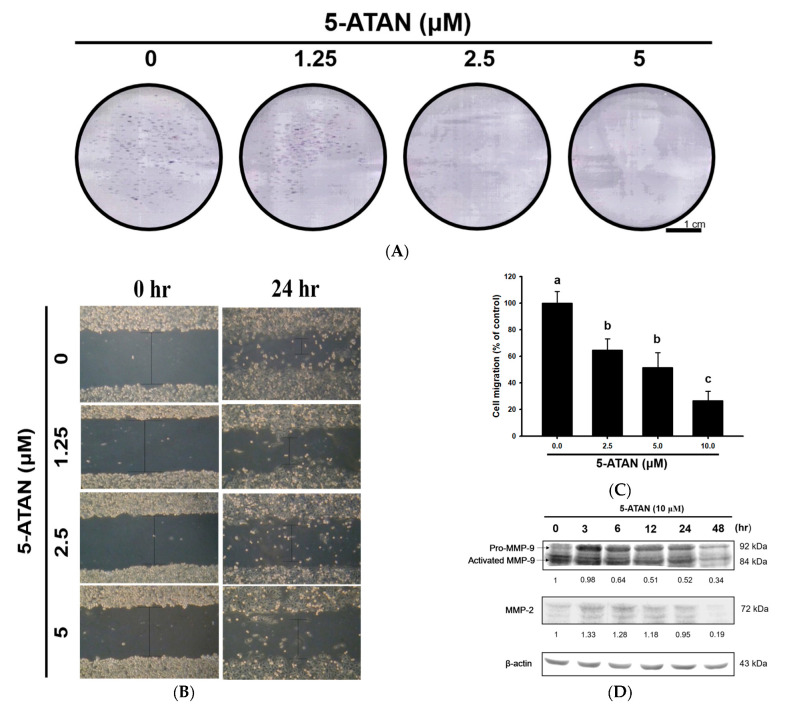
Effects of 5-ATAN on colony formation and cell migration in PC-3 cells. (**A**) Cells were treated with various concentrations of 5-ATAN, and the colonies formed were observed by crystal violet staining and photographed after 14 days of incubation. (**B**) Cell migration was analyzed by wound healing assay. Representative microphotographs of wound healing assay of PC-3 cells exposed to various concentrations of 5-ATAN at the beginning and end of incubation (0 and 24 h, respectively). (**C**) Quantification of cells that migrated into the wound is shown. Data were expressed as mean ± SD of three determinations presented in the bar graph and analyzed statistically using ANOVA and Duncan’s test. Different letters represent statistical differences among treatments, *p* < 0.05. (**D**) The protein expression of MMP-2 and MMP-9 was analyzed by Western blot.

**Figure 5 ijms-23-13284-f005:**
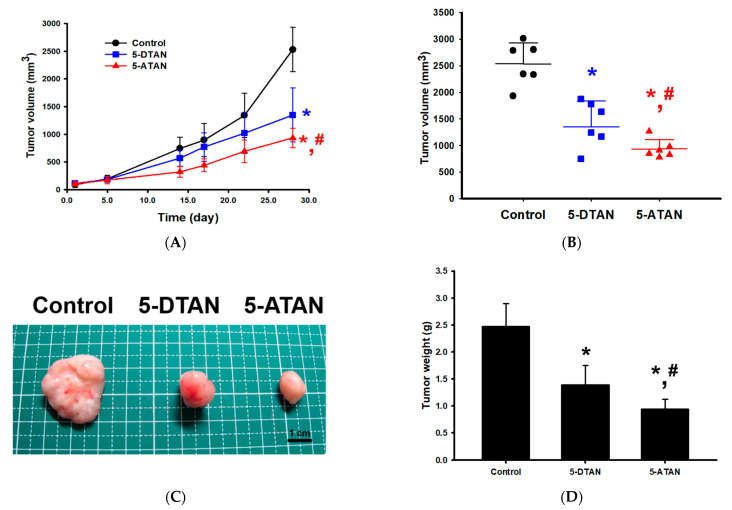
Effects of oral administration of 10 mg/kg 5-DTAN and 5-ATAN on the tumor growth of PC-3 cell xenografts. (**A**) The changes in tumor volume on the right flank side of nude mice as a function of time. (**B**) Tumor size at the end of the experiment. (**C**) The appearance of representative tumors harvested at the end of the experiment. (**D**) Mean tumor weight in nude mice. Data are expressed as means ± SD (*n* = 6). * *p* < 0.05 compared with control group. # *p* < 0.05 compared with 5-DTAN group.

**Figure 6 ijms-23-13284-f006:**
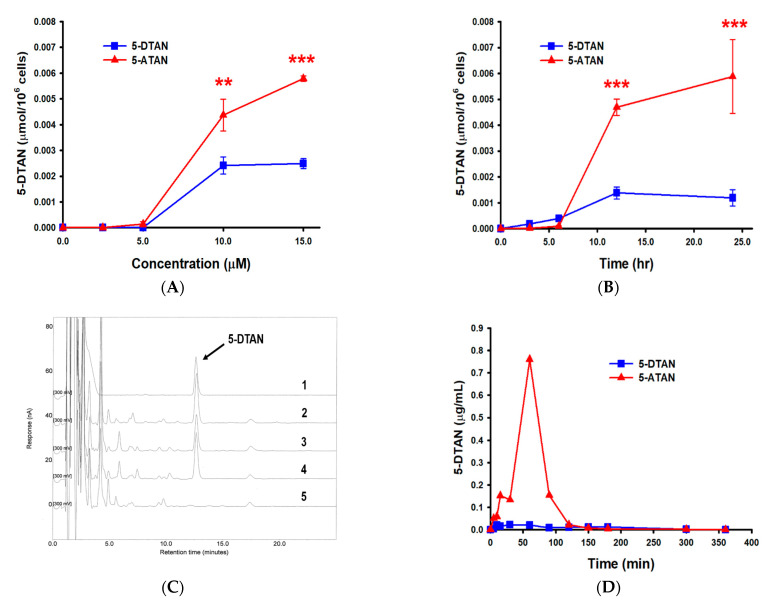
Cellular uptake and oral bioavailability of 5-DTAN. (**A**) Cytosolic concentrations of 5-DTAN in PC-3 cells treated with various concentrations of 5-DTAN or 5-ATAN for 12 h, (**B**) with 15 µM of 5-DTAN or 5-ATAN for different time periods. Data are expressed as means ± SD (*n* = 3). ** *p* < 0.01 and *** *p* < 0.001 compared with 5-DTAN-treated group. (**C**) Representative HPLC chromatograms of standard stock (1), plasma after administration of 5-ATAN (2–4), and blank plasma (5). (**D**) Mean plasma concentration-time curves of 5-DTAN after oral administration of 50 mg/kg of 5-DTAN or 5-ATAN to C57BL/6J mice.

**Figure 7 ijms-23-13284-f007:**
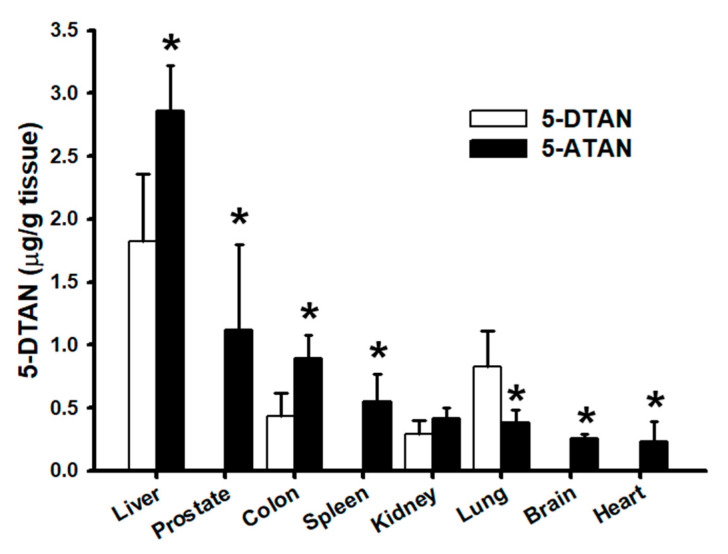
Tissue distribution of 5-DTAN in C57BL/6J mice following oral administration of 50 mg/kg of 5-DTAN or 5-ATAN for 7 consecutive days. Data are expressed as means ± SD (*n* = 6). * *p* < 0.05 compared with 5-DTAN-administered group.

**Table 1 ijms-23-13284-t001:** Effects of 5-DTAN and 5-ATAN on cell cycle distribution in PC-3 cells.

	Concentration (μM)	Cell Cycle Distribution (%)		
		G0/G1	S	G2/M
5-DTAN	0	63.8 ± 1.7 ^a^	8.5 ± 0.6	27.7 ± 1.3 ^b^
	2.5	60.2 ± 2.8 ^b^	8.2 ± 0.1	31.6 ± 2.7 ^a^
	5	59.9 ± 0.4 ^b^	8.7 ± 0.3	31.4 ± 0.5 ^a^
	10	58.2 ± 2.4 ^b^	8.5 ± 0.9	32.6 ± 1.1 ^a^
5-ATAN	0	55.3 ± 0.6 ^AB^	12.6 ± 0.4 ^A^	32.2 ± 0.9 ^D^
	2.5	58.5 ± 1.1 ^A^	6.8 ± 0.3 ^B^	34.7 ± 0.8 ^C^
	5	49.1 ± 0.3 ^B^	5.2 ± 0.3 ^C^	45.7 ± 0.5 ^B^
	10	44.7 ± 0.3 ^C^	4.6 ± 0.5 ^D^	50.3 ± 1.3 ^A^

Cells were treated with 0, 2.5, 5, and 10 µM of 5-DTAN or 5-ATAN for 48 h. Data are expressed as mean ± SD of three determinations and analyzed statistically using ANOVA and Duncan’s test. Different letters represent statistical differences in the same column among treatments, *p* < 0.05.

**Table 2 ijms-23-13284-t002:** Pharmacokinetic parameters of 5-DTAN in plasma of C57BL/6J mice following an oral single dose of 50 mg/kg of 5-DTAN or 5-ATAN.

Parameters	5-DTAN in Plasma	
	5-DTAN (50 mg/kg bw)	5-ATAN (50 mg/kg bw)
C_max_ (μg/mL)	0.030 ± 0.009	0.773 ± 0.196 *
T_max_ (min)	29.1 ± 17.1	59.7 ± 0.1 *
t_1/2_ (min)	173.4 ± 120.3	33.4 ± 9.8 *
AUC (μg/mL · min)	3.15 ± 0.24	34.73 ± 10.08 *

Data are expressed as mean ± SD (*n* = 6) and analyzed statistically using Student’s t test. * *p* < 0.05 compared with 5-DTAN group.

## Data Availability

Not applicable.

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
