# Peer review of "Acetylation Enhances the Anticancer Activity and Oral Bioavailability of 5-Demethyltangeretin"

_ijms, 2022, doi:10.3390/ijms232113284_

Round 1
Reviewer 1 Report
The manuscript " Acetylation Enhances the Anticancer Activity and
Oral Bioavailability of 5-Demethyltangeretin" researched the anticancer activity and oral bioavailability of the 5-acetylated TAN (5-ATAN). 5-ATAN increased the cytotoxicity of 5-DTAN against PC-3 human prostate
cancer cells and exhibited higher maximum concentration (Cmax) and area under the curve (AUC) of 5-DTAN in plasma. The overall experimental design of this manuscript is complete. However, there are some issues in the article that need to be revised.
I suggest that this article be carefully revised.
1. The graph of IC50 in Figure 1B is not standardized, please replace the analysis method and recalculate it.
2. The graphs of the results in the article are incomplete in many places, please replace them.
3. Please standardize the significance ANOVA markers throughout the article.
4. Please add the result containing the tumor size of each mouse.
5. How to determine 10 mg/kg as the optimal dosing concentration?
6. Since this manuscript examined the anticancer activity of 5-DTAN and 5-ATAN, why was PC-13 chosen in vitro instead of being guided by the results of tissue distribution experiments in vivo?
7. If 5-ATAN shortens the half-life of 5-DTAN, does achieving the same efficacy require an increase in the number of times the 5-ATAN is used?
Author Response
Comments:
The manuscript " Acetylation Enhances the Anticancer Activity and Oral Bioavailability of 5-Demethyltangeretin" researched the anticancer activity and oral bioavailability of the 5-acetylated TAN (5-ATAN). 5-ATAN increased the cytotoxicity of 5-DTAN against PC-3 human prostate cancer cells and exhibited higher maximum concentration (Cmax) and area under the curve (AUC) of 5-DTAN in plasma. The overall experimental design of this manuscript is complete. However, there are some issues in the article that need to be revised.
I suggest that this article be carefully revised.
- The graph of IC50 in Figure 1B is not standardized, please replace the analysis method and recalculate it.
Response: Thanks for reviewer's suggestion. We have replaced the analysis method and recalculate it, and the IC50 values of TAN, 5-DTAN, and 5-ATAN were 17.2 µM, 11.8 µM, and 5.1 µM, respectively. We have made revisions in the article (The sections of Abstract and Results and Discussion).
- The graphs of the results in the article are incomplete in many places, please replace them.
Response: Thanks for reviewer's suggestion. This situation may be due to the format changed when uploading the manuscript. We have replaced or adjusted the graphs, including Figures 1, 2, 4, and 6.
- Please standardize the significance ANOVA markers throughout the article.
Response: Thanks for reviewer's suggestion. We have revised statistical statement in the section of Statistical Analysis that statistical analysis of in vitro studies, comparisons of statistical significance between groups were determined by one-way analysis of variance (ANOVA) with a Duncan’s test. Statistical analysis of in vivo studies, comparisons of statistically significant differences among groups were made by the paired Student's t test. A p<0.05 was considered statistically significant.
- Please add the result containing the tumor size of each mouse.
Response: Thanks for reviewer's suggestion. We have added the result containing the tumor size of each mouse in Figure 5B.
- How to determine 10 mg/kg as the optimal dosing concentration?
Response: Thanks for reviewer's asking. We conducted a preliminary experiment before the formal experiment and found that the dose of 10 mg/kg had a significant inhibitory effect on tumor growth. In the future, we will further investigate the effects of different doses of 5-ATAN on tumor growth in xenograft model.
- Since this manuscript examined the anticancer activity of 5-DTAN and 5-ATAN, why was PC-13 chosen in vitro instead of being guided by the results of tissue distribution experiments in vivo?
Response: Thanks for reviewer's question. First of all, in order to understand whether acetylation can increase the anticancer activity of 5-DTAN, we selected PC-3 prostate cancer cell line for in vivo and in vitro experiments. The results confirmed that acetylation indeed enhanced the growth inhibitory effects of 5-DTAN on PC-3 cells. Next, we chose a model commonly used to study pharmacokinetics or tissue distribution, which is to use C57BL/6 mice to investigate whether acetylation could increase the tissue distribution of 5-DTAN in animals. Our results also confirmed that acetylation improved the tissue distribution of 5-DTAN in C57BL/6 mice, especially in several organs or tissues, including the prostate. In the future, the animal model of prostate carcinoma in situ can be used to investigate the inhibitory activity of 5-ATAN on the tumor growth.
- If 5-ATAN shortens the half-life of 5-DTAN, does achieving the same efficacy require an increase in the number of times the 5-ATAN is used?
Response: Thanks for reviewer's question. Although 5-ATAN shortened the half-life of 5-DTAN, administration of 5-ATAN increased the AUC of 5-DTAN by about 11 times than that of administration of 5-DTAN. Therefore, it is not necessary to increase the number of uses of 5-ATAN to achieve the same efficacy. It can be proved from the results of tissue distribution in mice that under the same administration times, administration of 5-ATAN significantly increased the content of 5-DTAN in most tissues.

Reviewer 2 Report
The manuscript entitled Acetylation Enhances the Anticancer Activity and Oral Bioavailability of 5-Demethyltangeretin discusses the influence of acetylation on anticancer activity and oral bioavailability of 5-DTAN in mice and prostate cancer cells. The results are well presented, but the connection between prostate cells and mice is not particularly highlighted. Minor corrections are needed before the article is published:
- the summary is too long and almost loses its meaning
- there are many extra spaces in the text and the text needs to be proofread
- confusion at the abbreviations MMP used, which are the designation for mitochondrial membrane potential and, on the other hand, matrix metalloproteinase
- images need processing: poor resolution, poor statistical data labels, the text in Figures is too small, missing text
- Table 1: all decimal places must be the same
- Figure 6 C: HPLC chromatograms should be numbered as suggested in the image title.
- The description of the HPLC method is missing in the experimental part, as the chromatograms were presented on the Figure 6.
After taking into account all comments, the article will be suitable for publication in this journal.
Author Response
Comments:
The manuscript entitled Acetylation Enhances the Anticancer Activity and Oral Bioavailability of 5-Demethyltangeretin discusses the influence of acetylation on anticancer activity and oral bioavailability of 5-DTAN in mice and prostate cancer cells. The results are well presented, but the connection between prostate cells and mice is not particularly highlighted. Minor corrections are needed before the article is published:
- The summary is too long and almost loses its meaning.
Response: Thanks for reviewer's suggestion. We have condensed the Abstract section of this manuscript.
- There are many extra spaces in the text and the text needs to be proofread.
Response: Thanks for reviewer's suggestion. We have proofread the text and deleted certain extra spaces. In addition, some appear to be extra spaces, probably caused by the justified (left-right alignment) chosen for the text's format.
- Confusion at the abbreviations MMP used, which are the designation for mitochondrial membrane potential and, on the other hand, matrix metalloproteinase.
Response: Thanks for reviewer's reminder. We have deleted the abbreviation for mitochondrial membrane potential. The MMP abbreviation used in the article now refers to the matrix metalloproteinase.
- Images need processing: poor resolution, poor statistical data labels, the text in Figures is too small, missing text.
Response: Thanks for reviewer's suggestion. This situation may be due to the format changed when uploading the manuscript. We have replaced or adjusted the graphs, including Figures 1, 2, 4, and 6.
- Table 1: all decimal places must be the same.
Response: Thanks for reviewer's suggestion. We have corrected the decimal places in table 1.
- Figure 6 C: HPLC chromatograms should be numbered as suggested in the image title.
Response: Thanks for reviewer's suggestion. This situation may be due to the format changed when uploading the manuscript. We have adjusted the graphs and the HPLC chromatograms has numbered as suggested in the image title.
- The description of the HPLC method is missing in the experimental part, as the chromatograms were presented on the Figure 6.
Response: Thanks for reviewer's suggestion. We have addressed the HPLC method to the section of Materials and Methods (3.9.) in the manuscript.

Round 2
Reviewer 1 Report
The revised version of the manuscript could be accetped.